# Interpretation of continuously measured vital signs data of COVID-19 patients by nurses and physicians at the general ward: A mixed methods study

**Harriët M. R. van Goor**[1]*, **Martine J. M. Breteler**[1,2,3], **Lisette Schoonhoven**[4], **Cor J. Kalkman**[2], **Kim van Loon**[2], **Karin A. H. Kaasjager**[1]

**1** Department of Acute Internal Medicine, University Medical Center Utrecht, Utrecht, The Netherlands, **2** Department of Anesthesiology, University Medical Center Utrecht, Utrecht, The Netherlands, **3** Department of Digital Health, University Medical Center Utrecht, Utrecht, The Netherlands, **4** Department of Public Health, Healthcare Innovation & Evaluation and Medical Humanities, Julius Center, University Medical Center Utrecht, Utrecht, The Netherlands

* h.m.r.vangoor-3@umcutrecht.nl

## Abstract

### Background

Continuous monitoring of vital signs is introduced at general hospital wards to detect patient deterioration. Interpretation and response currently rely on experience and expert opinion. This study aims to determine whether consensus exist among hospital professionals regarding the interpretation of vital signs of COVID-19 patients. In addition, we assessed the ability to recognise respiratory insufficiency and evaluated the interpretation process.

### Methods

We performed a mixed methods study including 24 hospital professionals (6 nurses, 6 junior physicians, 6 internal medicine specialists, 6 ICU nurses). Each participant was presented with 20 cases of COVID-19 patients, including 4 or 8 hours of continuously measured vital signs data. Participants estimated the patient's situation ('improving', 'stable', or 'deteriorating') and the possibility of developing respiratory insufficiency. Subsequently, a semi-structured interview was held focussing on the interpretation process. Consensus was assessed using Krippendorff's alpha. For the estimation of respiratory insufficiency, we calculated the mean positive/negative predictive value. Interviews were analysed using inductive thematic analysis.

### Results

We found no consensus regarding the patient's situation (α 0.41, 95%CI 0.29–0.52). The mean positive predictive value for respiratory insufficiency was high (0.91, 95%CI 0.86–0.97), but the negative predictive value was 0.66 (95%CI 0.44–0.88). In the interviews, two themes regarding the interpretation process emerged. "Interpretation of deviations" included the strategies participants use to determine stability, focused on finding deviations in data.

**Data Availability Statement:** Anonymized answers to the questionnaire are summarized in supplementary table 1. Supplementary material 2 and 3 contain the anonymized dataset and R script used for the quantitative analysis. The baseline information of the 24 participants will not be publicly shared due to their privacy sensitive nature, but can be shared upon reasonable request to the corresponding author.

**Funding:** The author(s) received no specific funding for this work.

**Competing interests:** The authors have declared that no competing interests exist.

"Inability to see the patient" entailed the need of hospital professionals to perform a patient evaluation when estimating a patient's situation.

## Conclusion

The interpretation of continuously measured vital signs by hospital professionals, and recognition of respiratory insufficiency using these data, is variable, which might be the result of different interpretation strategies, uncertainty regarding deviations, and not being able to see the patient. Protocols and training could help to uniform interpretation, but decision support systems might be necessary to find signs of deterioration that might otherwise go unnoticed.

## Introduction

Continuous wireless monitoring of vital signs is introduced at general hospital wards for it's potential to observe vital (in)stability in time and remote [1]. During the COVID-19 pandemic, continuous monitoring gained even more interest since hospitals had to deal with a high number of severely ill patients who were cared for in isolation units. Although a recent observational study showed a reduction in intensive care unit (ICU) admissions after introduction of continuous monitoring [2], few randomised trials have been performed, and systematic reviews failed to show an unequivocal positive effect on patient outcomes [1, 3]. Continuous monitoring is a multifactorial intervention that includes multiple technical, organisational and behavioural challenges. Many of the non-technical issues are scarcely researched and might benefit from a more structured approach to implementation. One of the important factors of implementation is training of hospital staff [4, 5]. Previous studies show that nurses perceived an increased benefit of continuous monitoring if they were trained and confident in its use [6, 7]. In addition, hospital professionals experience a learning curve over time as they grow more acquainted with the system [5]. Still, many nurses feel like they have insufficient knowledge to interpret vital signs trends, and are unsure which deviations are important to recognise [5]. No guidelines or clinical scores (such as the Early Warning Score for intermittently measured vital signs data) currently exist to assist in the interpretation of continuously monitored vital signs data at the general ward. Practice therefore relies on experience and expert opinion.

In this study we aimed to gain understanding of how nurses and physicians, with and without training and experience, handle continuously measured vital signs of COVID-19 patients. The main objective was to determine whether consensus exists among nurses and physicians on a patient's vital status, based on interpretation of continuously measured vital sign trends. Secondly, we assessed the ability of nurses and physicians to recognise which COVID-19 patients are becoming respiratory insufficient, using the continuous monitoring data. Lastly, we aimed to identify how the interpretation process of hospital professionals works, and identify aspects of interpretation that should be included in training.

## Methods

We performed a mixed methods study at the tertiary medical centre University Medical Centre Utrecht, The Netherlands, between April 6th and August 9th 2022. At the designated COVID-19 ward of the hospital a continuous monitoring system had been in place between April 2020 until March 2021, but no continuous monitoring system was in place during the study period. Participation consisted of two parts: a case review of 20 cases of COVID-19 patients, followed

by a semi-structured interview. Ethical review was waived by the local Medical Ethics Review Committee (MERC Utrecht 22–091).

## Participant inclusion

Three groups of hospital professionals from the internal medicine department (general ward nurses, junior physicians, medical specialists) were included to participate in the study. A fourth group consisting of ICU nurses was added to function as a comparison group, since ICU nurses are trained and experienced in continuously measuring vital signs. Participants were eligible if they had gained experience with COVID-19 patients care in the previous two years. Participants were recruited via a notice in the weekly newsletter, or they were approached by the research team directly. The research team aimed to include a selection of participants representative of the hospital staff with regard to baseline characteristics. Participants were asked for informed consent digitally and the following baseline information was collected: age, gender, function, years of clinical experience, and previous experience with continuous monitoring.

## Case review questionnaire

The case review consisted of 20 real-life cases that were included in a previously published study on continuous monitoring of COVID-19 patients (MERC UMC Utrecht, 20–365) [8]. Cases were randomly selected from a prospective cohort with 429 patients depending on the outcome variable "respiratory insufficiency". Ten cases (50%) were randomly selected from the cohort that developed respiratory insufficiency, the other ten cases from the cohort that did not develop respiratory insufficiency. Participants were unaware of this fifty-fifty case mix. Per outcome category, continuous data was alternately plotted over 4- and 8-hour time series. For patients who developed respiratory insufficiency, the time series ended within 2 hours before the occurrence of this endpoint. Respiratory insufficiency was defined as the need for 15L/min or more oxygen administration, mechanical ventilation, resuscitation or death, whichever came first. Cases were presented anonymously. For each case, a short summary including age, sex, relevant medical history, duration of hospitalization and list of relevant medications was given. Subsequently, four visual plots of continuously monitored data were shown: heart rate, respiratory rate, oxygen saturation and amount of administered oxygen. Per case, the participants had to answer four questions: 1. Do you consider the situation of the patient improving, stable or deteriorating? 2. Would you take action in this situation? 3. Do you expect this patient to become respiratory insufficient in the next few hours? 4. Would you score this case easy or difficult to interpret? The case review questionnaire was sent digitally to the participant within one day before the scheduled interview, using the data capture tool Castor version 2022.1 (Castor Electronic Data Capture (2022)). The questionnaire took approximately 30 minutes to finish.

## Semi-structured interview

The focus of the qualitative part of the study was to understand the underlying rationale for the answers participants had given during the case review. We aimed to understand how the interpretation process works and unravel aspects of continuously measured vital signs that influence interpretation. The semi-structured interview guide can be found in (S1 File). Interviews were held one on one, either in real life or via the video call using Microsoft Teams. The completed questionnaires of the participant was shown to refer to during the interview. The interviewer (HvG) had no previous experience with or training in scientific interviewing. She was coached and guided by experienced interviewers in the research group (LS, KvL, MB)

throughout the study. The interviewer was employed as PhD student during the study and participants knew her as a PhD student with focus on continuous monitoring of vital signs. In the study information it was stated that the interviewer wanted to 'gain insight into the decision making process'. Both the questionnaire and interview were pilot tested before the start of the study. During interviews, audio was recorded for transcription. No field notes were made. Transcripts were not returned to participants for comments.

## Sample size and quantitative analysis

The sample size was determined based on the first research question: Is there consensus amongst nurses and physicians over a patient's vital status based on continuously measured data? To determine the interobserver agreement we used Krippendorff's alpha coefficient, since this estimate allows for more than two answer options and more than two observers [9]. We aimed to determine the interobserver agreement for the entire sample and for the four hospital staff groups individually. Based on equal a-priori probability of the three outcome options (improving, stable, deteriorating) and a Krippendorff's alpha of at least 0.8 for sufficient agreement, and 0.67 for moderate agreement, 116 values were needed per group [10]. By using 20 cases per participant and 6 participants per group, we acquired 120 values per group. The confidence interval (CI) was calculated using a bootstrapping method with 2000 iterations. Secondly, we determined the interobserver agreement for 'action' and 'expected respiratory insufficiency'. For the second research question, we determined the positive and negative predictive value of the estimation of pending respiratory insufficiency by hospital staff per participant, and used these to calculate the mean positive and negative predictive value. A 95%CI was used for hypothesis testing. R software version 4.0.3 (R foundation for Statistical Computing (2021)) was used for quantitative analysis.

## Qualitative analysis

All interviews were transcribed using a naturalized verbatim style by HvG [11] and uploaded in data analysis software NVivo 12 (QRS International Pty Ltd. (2018)). Subsequently, we used inductive thematic content analysis strategy to find recurrent themes within the interviews [12]. The first two stages (familiarizing with data, generating the initial codes, and searching for themes) were performed by HvG and MB separately. Reviewing, defining and naming themes, and producing reports was done by HvG and MB together, and was hereafter discussed with LS. The study reported following the consolidated criteria for reporting qualitative research (COREQ) guidelines [13].

## Mixing methods

The quantitative and qualitative data were mixed in two ways. First of all, the qualitative part of the study (the interview) built on the quantitative questionnaire, and the answers to the questionnaire were used in the interview. Secondly, the results of the quantitative and qualitative part were merged in the discussion [14].

## Results

Of the 24 participants, 5 contacted the research team after the notice in the newsletter, and 19 were approached personally (Table 1). All participants had worked multiple shifts at COVID-19 units in the previous 2 years. All included nurses were female, whereas only 1 of the 6 medical specialists was female. Junior physicians had the least clinical experience (median of 2 years, range 1–4), medical specialists had the most experience (median 17.5 years, range 8–18).

**Table 1. Participant characteristics.**

|  | All (N = 24) | Nurses (N = 6) | Junior physicians (N = 6) | Medical specialists (N = 6) | ICU nurses (N = 6) |
|---|---|---|---|---|---|
| Age (median, range) | 31 (23–56) | 25 (22–29) | 28 (25–31) | 42.5 (35–45) | 31.5 (30–56) |
| Gender (N) |  |  |  |  |  |
| - Female | 14 (58%) | 6 (100%) | 3 (50%) | 1 (17%) | 4 (67%) |
| - Male | 10 (42%) | 0 (0%) | 3 (50%) | 5 (83%) | 2 (33%) |
| Recruitment (N) |  |  |  |  |  |
| - Newsletter | 5 (21%) | 2 (33%) | 0 (0%) | 0 (0%) | 3 (50%) |
| - Personally approached | 19 (79%) | 4 (67%) | 6 (100%) | 6 (100%) | 3 (50%) |
| Years of clinical experience (median, range) | 5.5 (1–28) | 3.75 (2.5–6) | 2 (1–4) | 17.5 (8–18) | 10.5 (5–28) |
| Experience with continuous monitoring (N) |  |  |  |  |  |
| - Previous experience | 8 (33%) | 0 (0%) | 1 (17%) | 1 (17%) | 6 (100%) |
| - Little or no experience | 16 (67%) | 6 (100%) | 5 (83%) | 5 (83%) | 0 (0%) |

All ICU nurses had experience with continuous monitoring during their daily work at the ICU. One medical specialist had experience with the continuous monitoring at the COVID-19 unit between 2020–2021, and one junior physician had assisted in another study on the use of a wearable sensor for continuous vital signs monitoring. The other participants had little or no experience with continuous monitoring outside the high care unit.

## Quantitative analysis

No agreement on the situation of the patient (improving, stable or deterioration) was found among hospital professionals, both in the overall analysis and the subgroup analyses (Table 2, S1 Table). No agreement was found on whether to take action. Agreement was not found on the expectation of respiratory insufficiency, but could also not be ruled out, with an upper limit of the 95%CI in the 'moderate agreement' range. For the prediction of respiratory insufficiency, the mean positive predictive value of all participants was 0.91 (95%CI 0.86–0.97). The mean negative predictive value was 0.66 (95%CI 0.44–0.88).

## Qualitative analysis

We performed an inductive analysis to find recurrent themes on how continuously measured vital signs at was interpreted by hospital professionals. Two main themes emerged: 'Interpretation of deviations' and 'Inability to see the patient'.

**Interpretation of deviations.** When assessing the vital sign data, participants were often focussed on finding deviations. Since the cases were all COVID-19 patients, deviations in respiratory rate and oxygen saturation received the most attention. Participants used one of three strategies to find these deviations: by assessing thresholds breaches (e.g. a respiratory rate above 30/min), by assessing trends (an increasing respiratory rate over time), or, most often, by using both thresholds and trends (a respiratory rate that increased over time and exceeds

**Table 2. Analysis of interrater agreement using Krippendorff's alpha coefficient (α).** Grades of agreement of α: <0.67 low agreement, 0.67–0.80 moderate agreement, >0.80 high agreement.

|  | All | Nurses | Junior physicians | Medical specialists | ICU nurses |
|---|---|---|---|---|---|
| Situation (α, 95%CI) | 0.41 (0.29–0.52) | 0.32 (0.198–0.45) | 0.46 (0.35–0.57) | 0.49 (0.37–0.60) | 0.44 (0.33–0.54) |
| Action (α, 95%CI) | 0.37 (0.20–0.54) | 0.32 (0.12–0.50) | 0.36 (0.16–0.54) | 0.43 (0.23–0.62) | 0.52 (0.34–0.68) |
| Respiratory insufficiency (α, 95%CI) | 0.59 (0.40–0.76) | 0.59 (0.41–0.76) | 0.65 (0.47–0.82) | 0.71 (0.52–0.87) | 0.52 (0.36–0.70) |

30/min). If no deviations were found, participants stopped the assessment, and the patient was deemed stable. These cases were usually considered easy. If deviations were constantly present and very clearly pointing in one direction, e.g., an oxygen saturation that progressively decreased over the entire plot from 98% to 88%, participants considered the case easy too. Very short deviations were often disregarded by participants as being measurement errors or motion artefacts. However, most participants experienced difficulty if deviations occurred irregularly. Participants felt uncertain how long or severe a deviation had to be to reflect true patient deterioration. Similarly, participants did not know how many hours of data were needed for a valid estimation of the patient's situation. Consequently, lack of knowledge of target values, periods of missing data, and fluctuating trends posed difficulties for participants. One participant said: *"..and for example the first 2 hours is stable, but the next 2 hours has fluctuations and after that it is stable again. Are you going to intervene, or accept it, or how should you interpret this"*. In these cases, participants tried to find more information or conformation in other sources, such as corresponding deviations in other vital signs, information on the time of day, or patient history. For example, obstructive sleep apnoea disease could be a logical explanation for nightly dips in oxygen saturation. If a satisfying explanation could be found here, participants were more confident to make a decision regarding the patient's status: *"It's likely that, there is an increased respiratory rate here, an increased heart rate, maybe she was doing personal care, or she went to the toilet. I see it's 9 o'clock in the evening, she was probably doing something"*. In cases when interpretation was considered difficult, participants often fell back on using the last vital sign values of the plot and comparing these against thresholds, since this is the method they currently use on intermittent vital signs: *"..you try to look at the trend, but you end up interpreting everything as a snapshot. As you are used to, that there is one measuring moment"*. Many participants did experience a learning curve during the study, gaining more confidence interpreting deviations after they had seen more cases. This learning curve was experienced by all hospital professionals, including ICU nurses. ICU nurses indicated that they were not used to looking at vital signs data this way either, and mostly rely on threshold alarms. They did indicate to use repeated threshold alarms as an alternative form of trend monitoring: the increase or decrease of a threshold alarm could be seen as a trend.

**Inability to see a patient.**   Of the presented context information, only the amount of oxygen therapy was used regularly, to determine the risk of a patient becoming respiratory insufficient. The remaining patient characteristics, such as age, gender, medication, and comorbidity, were often read, but seldom used. Participants indicated that what they really needed, instead of these patient characteristics, was a live patient evaluation. Especially in cases of possible patient deterioration, participants missed the possibility to examine the patient. To physically see the patient, and to hear what symptoms they experience, could help provide context to findings in the vital signs data: *"If I would see these [plots], I would first go to the patient to see how he is doing, what do I observe, and then verify, does that match with what the values tell me"*. Participants often tried to make an estimation of possible exhaustion of the patient, which was hard to determine based on vital sign values alone. Especially for patients with an 'unstable' starting situation, e.g., a lot of oxygen therapy, participants had trouble to determine whether a patient was near the edge of exhaustion. In contrast, two participants said to have little difficulty interpreting the data without seeing the patient. One of them said: *"I am surprised by with how much, how little, data you can still have an idea [of the patient's situation]"*.

## Discussion

In this study, we aimed to gain understanding of how nurses and physicians currently handle continuously measured vital signs data of COVID-19 patients, presented graphically as trend

plots. We found little agreement among hospital professionals on the interpretation of stability of a patient's situation and inconsistent ability to recognise respiratory insufficiency, which might be the result of variable interpretation strategies, uncertainty regarding deviations in continuously measured vital signs data, or not being able to see the patient.

The variability and uncertainty in interpretation by participants might be one of the underlying reasons for the lack of consensus regarding a patient's vital stability. Hospital professionals usually try to quantify findings in vital signs to be able to communicate them with other hospital professionals, prioritize workload, and make decisions regarding escalation of care [15, 16]. In this study, hospital professionals struggled to quantify findings of continuously measured data. They were most confident in pointing out threshold breaches, which they are used to when using intermitted monitoring, but struggled to quantify the time related aspect of continuously measured such as length and frequency. This might be due to the lack of knowledge or experience, but could also be because this process occurs more subconsciously. Hospital professionals do register changes over time, but do not yet have the quantifying language to express these findings. In a previous study on continuous monitoring, nurses indicated to use continuous monitoring to "..*make sure that, just from a glance, that nothing has changed*" [16]. Without guidelines about when the length or frequency of a deviation starts to be concerning, the estimation of patient stability based on continuously measured data is bound to be a grey area.

The observed lack of consensus regarding patient stability might complicate the collaboration between hospital professionals, a collaboration which is important for successful execution of a rapid response to deterioration. Therefore, more information is needed on which deviations are indicators of deterioration. This includes not only the height or depth of the deviation, but also the length, frequency, and context. A clinical decision support protocol on how to quantify continuously measured vital signs data, and what action should be taken if deviations are found, could uniform the interpretation by hospital professionals and empower them to communicate patient deterioration based on continuously measured vital signs.

The results of this study underline the difficulty to interpret continuously monitored vital signs data, while not being able to see the patient. Seeing and speaking to a patient is indisputably an important part of patient evaluation to collect information on signs and symptoms [17, 18]. Furthermore, a patient's symptoms can trigger the non-analytical thought process that results in a 'gut feeling', which is a valuable part of the decision making process of nurses and physicians [18–20]. Although some decisions could be made based on vital signs data alone, physical evaluation was most commonly mentioned as missing in our study by both nurses and physicians. When taking intermittent vital signs, patient observation usually precedes vital sign evaluation. Physical cues are often early signs of deterioration, and vital signs are then used to confirm or quantify these findings [18, 21, 22]. With continuous monitoring, this can be done similarly, by evaluating the patient first and then comparing findings with the vital signs data of the past hours. However, while vital signs data are now continuously available, physical examination is not. We therefore might have to reverse the order and use clues in vital signs to find those patients who need bedside examination.

Not being able to do a physical examination of the patient might be a challenge especially for interventions including a remote monitoring centre [23]. In these centres, staff (often nurses) are appointed to monitor patients remotely and alert the nurses and physicians at the ward if needed. For these professionals, knowledge of trend interpretation without patient assessment is of the utmost importance. However, a remote monitoring strategy that solely relies on vital signs data will not be maximally effective. Therefore, remote monitoring centres have found solutions to create a more complete remote patient evaluation. Virtual ways of assessing the patient have been introduced, including two-way video and audio connection,

and the remote monitoring centre is included in the multidisciplinary care for patients to have access to more context information [23].

Previous studies into learning trajectories of continuous monitoring described a learning curve to work with a new method to monitoring vital signs [5, 24]. Jones et al. [24] described five stages of a learning trajectory for (wired) telemonitoring, supervised by cardiac care unit (CCU) nurses. The first three stages involve technical training in the use of the monitoring system, similar to what current projects starting with continuous remote monitoring experience. During these stages, the hospital staff needs to gain confidence in the reliability of measurements. Currently, the trust in the validity of continuously measured data is lower than in the validity of intermittently measured vital signs data [16]. In our study too, the possibility of 'measurement errors' was often mentioned by participants. The last two stages of the described learning trajectory involve gaining knowledge about interpretation, which takes several education days, e-learnings, and learning on the job. The role of CCU staff is critical in this strategy, since they are the experts providing education, and are partly responsible for the monitoring of patients. When introducing continuous remote monitoring at the general ward, there is a clear need for such experts, who know how to interpret data and are confident to teach others. We assumed that ICU nurses, who work with continuous vital signs data on a daily basis, would be experts in interpreting trend data. However, the ICU nurses in our study indicated that looking at graphical trends of vital signs was new to them, and they usually focused more on alarm thresholds instead of trends. Consciously using trends of vital signs appears to be a very specific way of looking at data that is not commonly practiced at the ward, and therefore finding suitable experts and teachers might prove challenging.

Even though there is a wish to recognise deterioration earlier, vital sign abnormalities have to be obvious before hospital staff feel the need to intervene [25]. In this study, participants indicated that respiratory insufficiency is easily recognised when abnormalities are severe, but not if they are moderate or fluctuating. They were seldom wrong when indicating that a patient was becoming respiratory insufficient, but missed some cases that were less obvious. Nurses indicate that alarms are not needed when continuous data is regularly assessed [26] and might even lead to alarm fatigue [27]. However, alarms might be necessary to detect signs of deterioration that are currently missed by hospital professionals, provide decision support in case of moderate or uncertain deviations, or help to find those patients that need clinical assessment. We should aim to find an alarm strategy that provides useful decision support without rendering too many false alarms. For future research, it would be interesting to see if these cases are more easily recognised if hospital staff has more training and experience with continuous monitoring, and when real life patient assessment is readily available.

## Strengths and limitations

This study not only investigated to what extent the interpretation of vital signs is similar among hospital professionals, but also evaluated how this interpretation comes to be, and highlighted the difficulties that are encountered during interpretation. The study was based on real, unfiltered data to create the most realistic situation. Because of the heterogeneous case pool, the emphasis was on the continuously measured vital signs data instead of the underlying disease. Nonetheless, since the interpretation process is influenced by the admission diagnosis, findings are most applicable to COVID-19 cases. Although we tried to make the interpretation process as realistic as possible, the study was still limited by the artificial circumstances of estimating a patient's condition with only paper-based information. However, these circumstances do resemble the situation for remote monitoring by personnel that has no direct access

to the patient themselves. Because the cases were randomly selected, some cases of patients that experienced respiratory insufficiency yielded only very non-specific deviations, making it hard for hospital professionals to detect upcoming deterioration. It is not certain that a technological solution would have been able to detect deterioration in these cases either. The participants included in the study were all enthusiastic to participate in a study regarding continuous monitoring, which might have led to selection bias. Although the subgroups differed in baseline regarding age, sex, and years of clinical experience, they did reflect the working population in our hospital. We incorrectly assumed that ICU nurses would be used to assessing trend data. Other groups of professionals, such as ICU physicians or anaesthetists, might be more used to this specific way of handling vital signs data and might therefore have been a more suitable comparison group.

## Conclusion

Little agreement was found among hospital professionals regarding the estimation of patient stability based on continuously measured vital signs of COVID-19 patients. Differences might be the result of variable interpretation strategies, uncertainty regarding deviations in continuous monitoring data, and not being able to see the patient. Protocols and targeted training could help to unify the interpretation of continuously measured vital signs by hospital professionals. Decision support systems however might be necessary to detect cases of deterioration that are not easily recognised.

## Supporting information

**S1 Table. Number of participants giving a certain answer, per case.** Total number of participants is 24.
(DOCX)

**S1 File. Semi structured interview guide (translated from Dutch).**
(DOCX)

**S1 Data. Dataset of quantitative data of questionnaire answers.**
(XLSX)

**S1 Text. R script for reproducing quantitative analysis.**
(TXT)

## Acknowledgments

We would like to thank all participating nurses and physicians for the time and effort they put in to make this study possible.

## Author Contributions

**Conceptualization:** Harriët M. R. van Goor, Martine J. M. Breteler, Cor J. Kalkman, Kim van Loon, Karin A. H. Kaasjager.

**Data curation:** Harriët M. R. van Goor.

**Formal analysis:** Harriët M. R. van Goor, Martine J. M. Breteler.

**Investigation:** Harriët M. R. van Goor, Martine J. M. Breteler.

**Methodology:** Harriët M. R. van Goor, Martine J. M. Breteler, Lisette Schoonhoven.

**Supervision:** Lisette Schoonhoven, Cor J. Kalkman, Kim van Loon, Karin A. H. Kaasjager.

**Visualization:** Harriët M. R. van Goor.

**Writing – original draft:** Harriët M. R. van Goor.

**Writing – review & editing:** Martine J. M. Breteler, Lisette Schoonhoven, Cor J. Kalkman, Kim van Loon, Karin A. H. Kaasjager.

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
