## [Decision Letter · Decision Letter 0]

21 Feb 2023

PONE-D-22-33707Interpretation of continuously measured vital signs data of COVID-19 patients by nurses and physicians at the general ward: a mixed methods studyPLOS ONE

Dear Dr. van Goor,

Thank you for submitting your manuscript to PLOS ONE. After careful consideration, we feel that it has merit but does not fully meet PLOS ONE’s publication criteria as it currently stands. Therefore, we invite you to submit a revised version of the manuscript that addresses the points raised during the review process.

We look forward to receiving your revised manuscript.

Kind regards,

Sampson Twumasi-Ankrah, PHD

Academic Editor

PLOS ONE

Journal Requirements:

Reviewers' comments:

Reviewer's Responses to Questions

**Comments to the Author**

1. Is the manuscript technically sound, and do the data support the conclusions?

Reviewer #1: Yes

Reviewer #2: Yes

2. Has the statistical analysis been performed appropriately and rigorously? 

Reviewer #1: No

Reviewer #2: Yes

3. Have the authors made all data underlying the findings in their manuscript fully available?

Reviewer #1: Yes

Reviewer #2: Yes

4. Is the manuscript presented in an intelligible fashion and written in standard English?

Reviewer #1: Yes

Reviewer #2: Yes

5. Review Comments to the Author

Reviewer #1: Abstract;

The conclusion should speak directly to the three clearly stated aims of finding consensus regarding vital sign interpretation, assess ability to recognize respiratory insufficiency and evaluation of the interpretation process.

Semi-structured interview

Use of untrained interviewer for the interviews present with inherent limitations and authors are to indicate remedies for minimizing this effect.

Authors mention they aimed to have a representative study population from the hospital, this must be explicitly stated under the sample population.

Participant inclusion

The inclusion criteria of experience with COVID-19 need further information as the frequency and how recent are likely to influence responses

Results

Under quantitative analysis, authors need to provide a legend on categorizing agreement as low, moderate or high to aid readers appreciate why the choice of these words.

Under table 2, authors should indicate at least a general estimate of p value informing the non-agreements e.g. p values less than ………

No analysis on PPV in results section but estimates are stated under the results narration.

Discussion

First paragraph under discussion do not report on key findings of all three objectives. Line 258-263

Conclusion

See above on abstract

Minor Comments

Line 42 includes should be included

Line 43 focused and not focussed

Line 44 entailed and not entails

Line 282; a collaboration that is should be replaced with “which”

Line 287; authors should consider “uniform” instead of unify

Line 319; described instead of describe

Line 335; focused instead of focussed

Reviewer #2: The robust nature of the study with good description of the methods. However the sampling for the quantitative arm of the study did not come out well. The discussion could have been more detailed capturing every set objective.

6. PLOS authors have the option to publish the peer review history of their article (what does this mean?). If published, this will include your full peer review and any attached files.

Reviewer #1: No

Reviewer #2: No

---

## [Author Response · Author response to Decision Letter 0]

14 Mar 2023

Reviewer's Responses to Questions

First of all, let us thank you for your time and effort in reviewing our work. We are very grateful for the chance to improve our report. Secondly, we would like to inform you that during revision, we found a small error in one of the R-scripts: we found that instead of a lower cutoff of 0.025, we accidentally used 0.25, which impacts the lower limit of the 95% CI for Krippendorff’s alpha. We have corrected the script and corrected the values throughout the manuscript. We have thoroughly checked all conclusions drawn based on these values, but none of the conclusions was affected by the new lower limit of the 95%CI. 

Below you will find our point by point answer to the questions of the reviewers. 

Reviewer #1: 

Abstract

The conclusion should speak directly to the three clearly stated aims of finding consensus regarding vital sign interpretation, assess ability to recognize respiratory insufficiency and evaluation of the interpretation process.

Thank you for pointing this out. We have added the recognition of respiratory insufficiency as a separate finding to the conclusion. All three aims of the study are now (subsequently) included in the conclusion. 

Semi-structured interview

Use of untrained interviewer for the interviews present with inherent limitations and authors are to indicate remedies for minimizing this effect.

Although the interviewer did not undergo formal training, she was coached and guided by three members of the research group with experience in interviews studies, to limit the effect of inexperience. This included designing the interview together and thoroughly assessing the interview before it was held. We have added information about coaching to the methods section in line 137/138.

Authors mention they aimed to have a representative study population from the hospital, this must be explicitly stated under the sample population.

This statement is included in the methods section under ‘Participant inclusion’ in line 100-101.

Participant inclusion

The inclusion criteria of experience with COVID-19 need further information as the frequency and how recent are likely to influence responses

We did not include information as to the frequency and how recent the experience with COVID-19 patients was, since this was not part of our inclusion criteria (any experience was considered sufficient). However, since the study was performed in 2022, all experience was gained during the previous two years (the pandemic) and all participants turned out to have had extensive experience as a consequence of the high case load during the pandemic. We have added this information to the results. 

Results

Under quantitative analysis, authors need to provide a legend on categorizing agreement as low, moderate or high to aid readers appreciate why the choice of these words.

The categorization as stated in the methods was added to the title of Table 2. 

Under table 2, authors should indicate at least a general estimate of p value informing the non-agreements e.g. p values less than ………

We have considered the use of a p-value in this context extensively, however we do not find it to be added value. We will explain why. We have 3 types of results: those with an 95%CI for the Krippendorff’s alpha entirely below 0.67, those with a value below 0.67 but an upper 95%CI limit above 0.67, and those with a value above 0.76 but a lower limit below 0.67. For the first category, we can say with much certainty that the agreement is low, since 95% of all bootstrap results falls below 0.67. For the second category, the value is ‘non agreement’, but the true agreement might still be ‘moderate’ since the upper limit is above 0.67. However, we have no certainty about this. For the last category, the value indicates agreement, but the lower limit is below 0.67, so again we have no certainty about this. These last two categories can be interpreted as ‘non-significant’. The H1 hypothesis of this study was that there would be agreement among professionals regarding interpretation. Since none of these values have a confidence interval entirely above 0.67, we cannot prove this hypothesis, and therefore concluded that we could not find agreement among hospital professionals. 

Since the 95%CI gives more information about the amount of uncertainty around the agreement value than a p-value, we choose to report the 95%CI. And since all conclusions as described above can be drawn using only a 95%CI, we did not think a p-value would be of added value. 

To help interpret these findings, we have added a sentence to the results explaining the uncertainty of the respiratory insufficiency expectation results. We hope this aids in the interpretation of the 95%CI. 

No analysis on PPV in results section but estimates are stated under the results narration.

We do not quite understand the point being made in this comment. We have included all results of our analyses, as stated in the methods, in the results section. Quantitative analyses (agreement and positive/negative predicting value) are stated under the quantitative analysis subheading, the results of the interview study are reported under the qualitative analysis subheading. Results including a large number of values are clarified using tables (table 1 and 2). Since the PPV an NPV only have 2 values, these were not summarized in a table. Hopefully this clarifies our structure of the results section.

Discussion

First paragraph under discussion do not report on key findings of all three objectives. Line 258-263

The objective that was missing was the assessment of the ability to recognize respiratory insufficiency. We have added the findings of this objective to the first paragraph of the discussion.

Conclusion

See above on abstract

Minor Comments

Line 42 includes should be included

Line 43 focused and not focussed

Line 44 entailed and not entails

Line 282; a collaboration that is should be replaced with “which”

Line 287; authors should consider “uniform” instead of unify

Line 319; described instead of describe

Line 335; focused instead of focused

Thank you for pointing out these errors, we have corrected them throughout the manuscript. 

Reviewer #2: The robust nature of the study with good description of the methods. However the sampling for the quantitative arm of the study did not come out well. The discussion could have been more detailed capturing every set objective.

We thank you for the feedback and are happy to read that the reviewer finds our study robust with a good description of methods. We are a little uncertain whether the ‘sampling for the quantitative arm of the study’ includes the sample size of the study, or the sample selection, or both. We have tried to clarify certain point of the sample selection, hopefully improving the information regarding sampling. The discussion has been made to include every set objective in the first paragraph.

---

## [Editor Report · Decision Letter 1]

2 May 2023

PONE-D-22-33707R1Interpretation of continuously measured vital signs data of COVID-19 patients by nurses and physicians at the general ward: a mixed methods studyPLOS ONE

Dear Dr. van Goor,

Thank you for submitting your manuscript to PLOS ONE. After careful consideration, we feel that it has merit but does not fully meet PLOS ONE’s publication criteria as it currently stands. Therefore, we invite you to submit a revised version of the manuscript that addresses the points raised during the review process.

Kind regards,

Sampson Twumasi-Ankrah, PHD

Academic Editor

PLOS ONE

Journal Requirements:

Additional Editor Comments:

1) For Table 1, include percentages to the frequencies for all categorical variables. e.g. Gender (Female 14(....%); Male (10(....%)

2) Kindly check and correct "where" in line 208
---

## [Author Response · Author response to Decision Letter 1]

7 May 2023

Dear dr. Twumasi-Ankrah, 

Thank you again for revising our manuscript. We are happy to see the previous corrections have been accepted. We have corrected the two additional points made:

- We have added percentages to table 1

- We have corrected the word ‘where’ to ‘were’ twice between lines 203-210

Hopefully our work is now up to your publication standards. We hope to hear from you soon,

Sincerely,

On behalf of the research team,

Harriët van Goor

---

## [Editor Report · Decision Letter 2]

9 May 2023

Interpretation of continuously measured vital signs data of COVID-19 patients by nurses and physicians at the general ward: a mixed methods study

PONE-D-22-33707R2

Dear Dr. van Goor,

We’re pleased to inform you that your manuscript has been judged scientifically suitable for publication and will be formally accepted for publication once it meets all outstanding technical requirements.

Kind regards,

Sampson Twumasi-Ankrah, PHD

Academic Editor

PLOS ONE
---

## [Editor Report · Acceptance letter]

16 May 2023

PONE-D-22-33707R2 

Interpretation of continuously measured vital signs data of COVID-19 patients by nurses and physicians at the general ward: a mixed methods study 

Dear Dr. van Goor:

I'm pleased to inform you that your manuscript has been deemed suitable for publication in PLOS ONE. Congratulations! Your manuscript is now with our production department. 

Kind regards, 

on behalf of

Dr. Sampson Twumasi-Ankrah 

Academic Editor

PLOS ONE